# The Truncated Lindley Distribution with Applications in Astrophysics

**Lorenzo Zaninetti**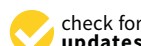

Physics Department, via P.Giuria 1, I-10125 Turin, Italy; zaninetti@ph.unito.it

**Abstract:** This paper reviews the Lindley distribution and then introduces the scale and the double truncation. The unknown parameters of the truncated Lindley distribution are evaluated with the maximum likelihood estimators. An application of the Lindley distribution with scale is done to the initial mass function for stars. The magnitude version of the Lindley distribution with scale is applied to the luminosity function for the Sloan Digital Sky Survey (SDSS) galaxies and to the photometric maximum of the 2MASS Redshift Survey (2MRS) galaxies. The truncated Lindley luminosity function allows to model the Malquist bias of the 2MRS galaxies.

**Keywords:** stars: normal; galaxy: groups, clusters, and superclusters; large scale structure of the Universe; Cosmology

## 1. Introduction

The Lindley distribution is defined by one parameter and was introduced to study the difference between fiducial distribution and posterior distribution, see [1,2]. Its detailed properties such as moments, cumulants, characteristic function, failure rate function and ... can be found in [3]. We now briefly outline some new trends, among others, for this distribution. A three parameter generalization of the Lindley distribution has been analyzed by [4], the truncated versions of the Lindley distribution has been studied by [5], and the estimation of the parameters of the generalized Lindley distribution has been done by [6] and a three-parameter Lindley distribution has been introduced by [7]. A careful analysis of these applications in the various fields of the natural sciences has revealed that the Lindley distribution has not yet been applied to astrophysics. Usually the mathematicians introduce many parameters, which characterize statistical distributions. In contrast, applications in the real world require fewer parameters, such as mean value and variance. The rapid development of computers has allowed to simulate the statistical distributions through the generation of random numbers, but this requires the evaluation of the inverse of the distribution function. A first example of an astrophysical application for a probability density function (PDF) is represented by the initial mass function for the stars (IMF). The distribution in mass of the stars has been fitted with a power law. This started with [8], who suggested that $\xi(m) \propto m^{-\alpha}$ where $\xi(m)$ represents the probability of having a mass between $m$ and $m + dm$; He found $\alpha = 2.35$ in the range $10\,M_\odot > M \geq 1\,M_\odot$. Subsequent research has started to analyze the initial mass function (IMF) with three power laws, see [9–11], and four power laws, see [12]. The approach to the IMF using a continuous distribution has been modeled by the lognormal distribution in order to fit both the range of the stars and the brown dwarfs (BDs) regime, see [13], by the beta distribution, see [14], by the truncated gamma distribution, see [15] and by the truncated lognormal distribution, see [16]. The previous analysis raises the following questions:

- Is it possible to find the constant of normalization for a left and right truncated Lindley PDF?
- Is it possible to derive an analytical expression for the mean of a left and right truncated Lindley PDF

- Is a left and right truncated Lindley PDF a model for the IMF and for a sample of masses?

A second example of an astrophysical application for a PDF is given by the luminosity function (LF) for galaxies. The Schechter function was the first LF for galaxies to be introduced, see [17]. Over the years, other LFs for galaxies have been suggested, such as a two-component Schechter-like LF, see [18], the hybrid Schechter+power-law LF to fit the faint end of the K-band, see [19], and the double Schechter LF, see [20]. To improve the flexibility at the bright end, a new parameter $\eta$ was introduced in the Schechter LF, see [21]. A third astrophysical application is in the photometric maximum visible in the number of cluster of galaxies as function of the redshift; for example, see Figure 7 in [22] where the number of galaxies as function of the redshift is plotted and Figure 2 in [23] where the number of clusters for three catalogs are reported as function of the redshift. The theoretical explanation of this effect is the joint distribution in redshift and and flux for galaxies; see formula (5.133) in [24] or formula (1.104) in [25] or formula (1.117) in [26]. Despite this theoretical background, the photometric maximum has been poorly analyzed. A fourth astrophysical application is in the range in absolute magnitude of galaxies versus the redshift visible in the various catalogs; for example, see Figure 9 in [22]. The mass of the stars in the IMF, the luminosity of galaxy in the LF and the absolute magnitude of galaxy in a given range of redshift vary between a minimum and a maximum value. This discussion suggests the introduction of finite boundaries for the Lindley IMF and LF rather than the usual zero and infinity following a pattern similar to the introduction of a left truncated beta LF; see [27], and for a left and right truncated Schechter LF luminosity function, see [28].

This paper reviews the original Lindley distribution in Section 2.1. It introduces the scaling in Section 2.2 and the double truncation in Section 2.3. The applications to the astrophysics are developed for the IMF, see Section 3, and for the luminosity function (LF) for galaxies, see Section 4.

## 2. The Lindley Family

We present a family of distributions of gradually increasing complexity.

### 2.1. Lindley Distribution

Let $X$ be a random variable defined in $[0, \infty]$; the *Lindley* probability density function (PDF), $f(x)$, is

$$f(x;c) = \frac{c^2 e^{-cx}(x+1)}{1+c},$$ (1)

the distribution function (DF), $F(x)$, is

$$F(x;c) = 1 - \left(1 + \frac{cx}{1+c}\right)e^{-cx},$$ (2)

where $c > 0$. At $x = 0$ $f(0) = \frac{c^2}{1+c}$ and not zero.

The average value or mean, $\mu$, is

$$\mu(c) = \frac{2+c}{c(1+c)},$$ (3)

the variance, $\sigma^2$, is

$$\sigma^2(c) = \frac{c^2 + 4c + 2}{c^2(1+c)^2}.$$ (4)

The $r$th moment about the origin and an approximation of the median are reported in Appendix A. The random generation of the Lindley variate X:c is given by

$$X:c \approx -\frac{W\left((R-1)(1+c)e^{-1-c}\right) + 1 + c}{c},$$ (5)

where $W$ is the Lambert W function, after [29], and R the unit rectangular variate R. The Lambert W function according to [30] is defined as

$$We^W = x. \tag{6}$$

The principal branch, $Wp(x)$, and the other branch, $Wm(x)$, of the Lambert W-function can evaluated with the Halley method

$$w_{n+1} = w_n - \frac{(w_n e^{w_n} - x)}{\left((w_n + 1) e^{w_n} - \frac{(w_n+2)(w_n e^{w_n} - x)}{2w_n+2}\right)}, \tag{7}$$

see [31–33]. The two branches of the Lambert W-function are reported in Figure 1.

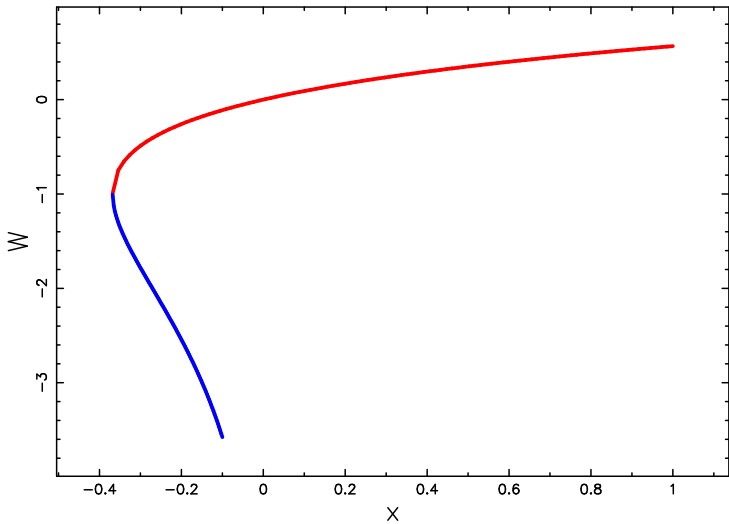

**Figure 1.** Wp(x) (red line) and Wm(x) (blue line).

A typical simulation of the Lindley PDF is reported in Figure 2.

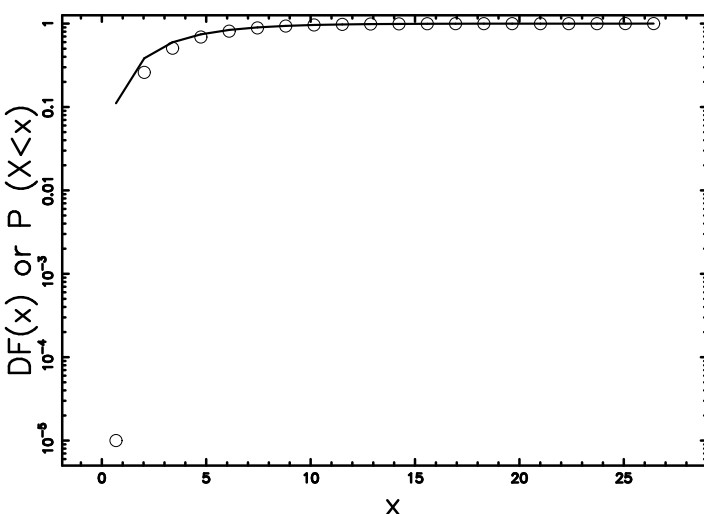

**Figure 2.** Histogram of the simulated Lindley PDF generated according to formula (5) and theoretical Lindley PDF (full line), 100,000 random points and $c = 0.5$.

The experimental sample consists of the data $x_i$ with $i$ varying between 1 and $n$; the sample mean, $\bar{x}$, is

$$\bar{x} = \frac{1}{n} \sum_{i=1}^{n} x_i, \tag{8}$$

the unbiased sample variance, $s^2$, is

$$s^2 = \frac{1}{n-1} \sum_{i=1}^{n} (x_i - \bar{x})^2, \tag{9}$$

and the sample $r$th moment about the origin, $\bar{x}_r$, is

$$\bar{x}_r = \frac{1}{n} \sum_{i=1}^{n} (x_i)^r. \tag{10}$$

The parameter $c$ can be obtained by the following match

$$\mu_1 = \bar{x}_1, \tag{11}$$

and therefore

$$\hat{c} = \frac{-\mu_1 + 1 + \sqrt{\mu_1{}^2 + 6\,\mu_1 + 1}}{2\,\mu_1}. \tag{12}$$

### 2.2. The Lindley Distribution with Scale

We now introduce the scale $b$ in the Lindley distribution and the PDF, $f_s(x; b, c)$, is

$$f_s(x; b, c) = \frac{c^2 e^{-\frac{cx}{b}} (x + b)}{b^2 (1 + c)}, \tag{13}$$

the DF, $F_s(x; b, c)$, is

$$F_s(x; b, c) = \frac{-e^{-\frac{cx}{b}} bc - e^{-\frac{cx}{b}} cx - e^{-\frac{cx}{b}} b + cb + b}{b (1 + c)}. \tag{14}$$

The mean, $\mu_s(c, b)$, is

$$\mu_s(c, b) = \frac{b(c + 2)}{c(1 + c)}, \tag{15}$$

and the variance, $\sigma_s^2(c, b)$, is

$$\sigma_s^2(c, b) = \frac{b^2 (c^2 + 4c + 2)}{c^2 (1 + c)^2}. \tag{16}$$

At $x = 0$ $f_s(0) = \frac{c^2}{(1+c)b}$.

The $r$th moment about the origin is reported in Appendix B.

The parameters $b$ and $c$ can be obtained by the following match

$$\mu_1 = \bar{x}_1 \tag{17a}$$
$$\sigma^2 = s^2, \tag{17b}$$

which means

$$\hat{c} = \frac{-3\,\bar{x}^3 + 3\,\bar{x}\,s^2 + \sqrt{2}\sqrt{(\bar{x}^2 - s^2)(2\,\bar{x}^2 - s^2)^2}}{\bar{x}^2 - s^2}, \tag{18}$$

and

$$\widehat{b} = \frac{1}{2} \frac{1}{\bar{x}\,(\bar{x}^2 - s^2)} \left( \sqrt{2}\sqrt{(\bar{x}^2 - s^2)\,(2\,\bar{x}^2 - s^2)^2} + (\bar{x}^2 - s^2) \left( \sqrt{2}\sqrt{\frac{s^2{}^2}{\bar{x}^2 - s^2}} - 4\,\bar{x} \right) \right). \tag{19}$$

The inequality $s^2 > \bar{x}^2/2$ makes both $\widehat{b}$ and $\widehat{c}$ negatives and, therefore, the sample is not suitable for a fit with Lindley distribution with scale.

### 2.3. The Truncated Lindley Distribution with Scale

Let $X$ be a random variable defined in $[x_l, x_u]$; the *truncated* PDF, $f_t(x; b, c, x_l, x_u)$, see [5,34], is

$$f_t(x; b, c, x_l, x_u) = \frac{f_s(x)}{F_s(x_u) - F_s(x_l)}, \tag{20}$$

and the DF, $F_t(x; b, c, x_l, x_u)$,

$$F_t(x; b, c, x_l, x_u) = \frac{F_s(x) - F_s(x_l)}{F_s(x_u) - F_s(x_l)}. \tag{21}$$

The inequality which fixes the range of existence is $\infty > x_u > x > x_l > 0$.
The first moment about the origin, $\mu'_{1,t}(b, c, x_l, x_u)$, is

$$\mu'_{1,t}(b, c, x_l, x_u) = \frac{MN1}{c\left( cbe^{\frac{cx_l}{b}} + cx_u e^{\frac{cx_l}{b}} - cbe^{\frac{cx_u}{b}} - cx_l e^{\frac{cx_u}{b}} + be^{\frac{cx_l}{b}} - be^{\frac{cx_u}{b}} \right)}, \tag{22}$$

where

$$MN1 = e^{\frac{cx_l}{b}} bc^2 x_u + e^{\frac{cx_l}{b}} c^2 x_u{}^2 - e^{\frac{cx_u}{b}} bc^2 x_l - e^{\frac{cx_u}{b}} c^2 x_l{}^2 + e^{\frac{cx_l}{b}} b^2 c + 2\,e^{\frac{cx_l}{b}} bcx_u - e^{\frac{cx_u}{b}} b^2 c$$
$$- 2\,e^{\frac{cx_u}{b}} bcx_l + 2\,e^{\frac{cx_l}{b}} b^2 - 2\,e^{\frac{cx_u}{b}} b^2, \tag{23}$$

and the second moment about the origin, $\mu'_{2,t}(b, c, x_l, x_u)$, is

$$\mu'_{2,t}(b, c, x_l, x_u) = \frac{MN2}{c^2\left( cbe^{\frac{cx_l}{b}} + cx_u e^{\frac{cx_l}{b}} - cbe^{\frac{cx_u}{b}} - cx_l e^{\frac{cx_u}{b}} + be^{\frac{cx_l}{b}} - be^{\frac{cx_u}{b}} \right)}, \tag{24}$$

$$MN2 = e^{\frac{cx_l}{b}} bc^3 x_u{}^2 + e^{\frac{cx_l}{b}} c^3 x_u{}^3 - e^{\frac{cx_u}{b}} bc^3 x_l{}^2 - e^{\frac{cx_u}{b}} c^3 x_l{}^3 + 2\,e^{\frac{cx_l}{b}} b^2 c^2 x_u$$
$$+ 3\,e^{\frac{cx_l}{b}} bc^2 x_u{}^2 - 2\,e^{\frac{cx_u}{b}} b^2 c^2 x_l - 3\,e^{\frac{cx_u}{b}} bc^2 x_l{}^2 + 2\,e^{\frac{cx_l}{b}} b^3 c + 6\,e^{\frac{cx_l}{b}} b^2 cx_u$$
$$- 2\,e^{\frac{cx_u}{b}} b^3 c - 6\,e^{\frac{cx_u}{b}} b^2 cx_l + 6\,e^{\frac{cx_l}{b}} b^3 - 6\,e^{\frac{cx_u}{b}} b^3. \tag{25}$$

The variance, $\sigma_t^2(b, c, x_l, x_u)$, is evaluated as

$$\sigma_t^2(b, c, x_l, x_u) = \mu'_{2,t} - (\mu'_{1,t})^2. \tag{26}$$

The parameters $b$ and $c$ can be evaluated with the maximum likelihood estimators (MLE), see Appendix C.

## 3. The IMF for Stars

The IMF for stars is actually fitted with three and four power laws, see [35,36]. The piece-wise broken inverse power law IMF is

$$p(m) \propto m^{-\alpha_i}, \tag{27}$$

each zone being characterized by a different exponent $\alpha_i$ and two boundaries $m_i$ and $m_{i+1}$. To have a PDF normalized to unity, one must have

$$\sum_{i=1,n} \int_{m_i}^{m_{i+1}} c_i m^{-\alpha_i} dm = 1. \tag{28}$$

The number of parameters to be found from the considered sample for the *n*-piece-wise IMF is $2n - 1$ when $m_1$ and $m_{n+1}$ are the minimum and maximum of the masses of the sample. In the case of $n = 4$, which fits also the region of brown dwarfs (BD), see [14], the number of parameters is seven. In the field of statistical distributions, the PDF is usually defined by two parameters. Examples of two-parameter PDFs are: the beta, gamma, normal, and lognormal distributions, see [37]. The lognormal distribution is widely used to model the IMF for the stars, see [13,38–40]. The lognormal distribution is defined in the range of $\mathcal{M} \in (0, \infty)$ where $\mathcal{M}$ is the mass of the star. Nevertheless, the stars have minimum and maximum values. In an example from the MAIN SEQUENCE, an M8 star has $\mathcal{M} = 0.06 \, \mathcal{M}_\odot$ and an O3 star has $\mathcal{M} = 120 \, \mathcal{M}_\odot$, see [41]. The presence of boundaries for the stars makes the analysis of the truncated lognormal, see [16], and of the truncated Lindley PDF attractive. In the case of the truncated Lindley PDF, the analysis of the samples representative of the IMF for stars is limited to those that produce both parameters $b$ and $c$ positive, and are therefore suitable for a fit with the truncated Lindley distribution. The statistical parameters are the same of [16] and are the merit function $\chi^2$, the reduced merit function $\chi^2_{red}$, the Akaike information criterion (AIC), the number of degrees of freedom $NF = n - k$ where $n$ is the number of bins and $k$ is the number of parameters, the goodness of the fit expressed by the probability $Q$, the maximum distance, $D$, between the theoretical and the astronomical DF and the significance level, $P_{KS}$, for the Kolmogorov–Smirnov test (K–S).

To give an example, Figure 3 reports the truncated Lindley DF for NGC 6611 with statistical parameters as in Table 1.

**Table 1.** Statistical parameters of NGC 6611 (207 stars + BDs) in the case of the truncated Lindley distribution. The number of linear bins, $n$, is 20.

| PDF | Method | Parameters | AIC | $\chi^2_{red}$ | $Q$ | $D$ | $P_{KS}$ |
|---|---|---|---|---|---|---|---|
| truncated Lindley | MLE | $b = 0.666$, $c = 1.938$, $x_l = 0.0189$, $x_u = 1.46$ | 47.75 | 2.48 | $8.4 \times 10^{-4}$ | 0.065 | 0.332 |
| lognormal | MLE | $\sigma = 1.029$, $m = 0.284$ | 71.24 | 3.73 | $1.3 \times 10^{-7}$ | 0.09366 | 0.04959 |
| truncated lognormal | MLE | $\sigma = 1.499$, $m = 0.478$, $x_l = 0.0189$, $x_u = 1.46$ | 50.96 | 2.68 | $2.8 \times 10^{-4}$ | 0.0654 | 0.372 |

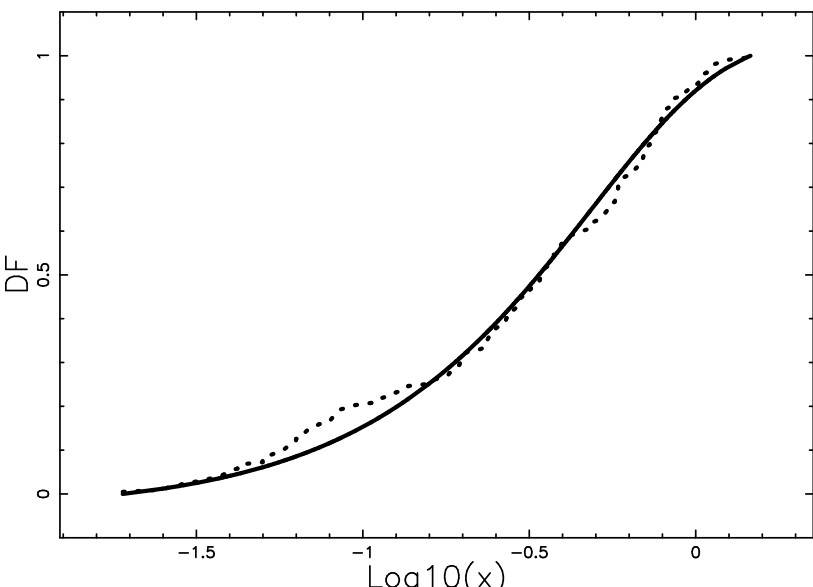

**Figure 3.** Empirical DF of mass distribution for NGC 6611 cluster data (207 stars + BDs) when the number of bins, *n*, is 20 (dotted points ) with a superposition of the truncated Lindley DF (full line). Theoretical parameters as in Table 1, MLE method. The horizontal axis has a logarithmic scale.

A careful analysis of Table 1 allows to conclude that in the case of NGC 6611 the truncated Lindley PDF produces a better fit in respect to the lognormal and truncated lognormal PDFs.

The lifetime of a star belonging to the MAIN V, $t_{MS}$, is

$$\frac{t_{MS}}{t_\odot} \approx \left(\frac{\mathcal{M}}{\mathcal{M}_\odot}\right)^{-2.5},$$

(29)

where $t_\odot$ is the lifetime of the sun, $10^{10}$ years, $\mathcal{M}$ is the mass of MAIN V star and $\mathcal{M}_\odot$ the solar mass, see http://astronomy.swin.edu.au/cosmos/ for more details. Figure 4 reports the modifications of the Lindley PDF with an increasing upper boundary. Meanwhile, Table 2 reports the correspondence between the selected mass and the connected lifetime.

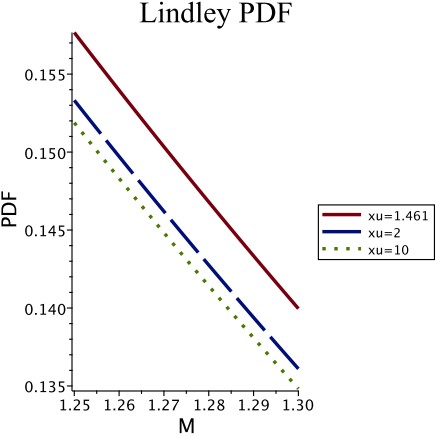

**Figure 4.** Double truncated Lindley PDF with parameters as in Table 1 and variable $x_u$; $x_u = 1.461$ (red full line), $x_u = 2$ (blue dashed line) and $x_u = 10$ (green dotted line).

**Table 2.** Lifetime of MAIN V star.

| Mass in Solar Units | Lifetime (Year) |
| --- | --- |
| 1.461 | $3.87 \times 10^9$ |
| 2 | $1.76 \times 10^9$ |
| 10 | $3.16 \times 10^7$ |

For example, in the case of the cluster NGC 6611, the upper limit in mass will decrease from $1.4\,M_\odot$ to $1\,M_\odot$ in $9.9\,10^9$ years and after that time the total number of stars will be the 92.25% of the original number of stars. The above model allows to see how the time modifies the Hertzsprung-Russell (H-R) diagram, i.e., the $M_V$ against $(B - V)$, in the young clusters of stars.

## 4. The Luminosity Function for Galaxies

In this section, we review the standard luminosity function (LF) for galaxies, we introduce a Lindley LF and a truncated Lindley LF, we then outline the formulae of the photometric maximum and we parametrize the averaged absolute magnitude as function of the redshift.

### 4.1. The Schechter Function

The Schechter function, introduced by [17], provides a useful fit for the LF of galaxies

$$\Phi(L; \alpha, L^*, \Phi^*)dL = \left(\frac{\Phi^*}{L^*}\right)\left(\frac{L}{L^*}\right)^\alpha \exp\left(-\frac{L}{L^*}\right)dL,$$

(30)

here $\alpha$ sets the slope for low values of $L$, $L^*$ is the characteristic luminosity and $\Phi^*$ is the normalization. The equivalent distribution in absolute magnitude is

$$\Phi(M)dM = 0.921\Phi^* 10^{0.4(\alpha+1)(M^*-M)} \exp\left(-10^{0.4(M^*-M)}\right)dM, \tag{31}$$

where $M^*$ is the characteristic magnitude as derived from the data. We now introduce the parameter $h$ which is $H_0/100$, where $H_0$ is the Hubble constant. The scaling with $h$ is $M^* - 5\log_{10} h$ and $\Phi^* h^3$ [Mpc$^{-3}$]. The numerical exploration of a new LF for galaxies requires that the $\chi^2_{red}$ is smaller or approximately equal to that of the Schechter LF. As an example, the LF given by the generalized gamma distribution with four parameters gives $\chi^2_{red}$ smaller than that of the Schechter LF in the five bands of SDSS galaxies, see Equation (21) an Table II in [42]

*4.2. The Lindley LF*

We start with the Lindley PDF with scaling as given by Equation (13),

$$\Psi(L; c, L^*, \Psi^*)dL = \frac{\Psi^* c^2 e^{-\frac{cL}{L^*}} (L+L^*)}{L^{*2} (1+c) \, dL} \, dM, \tag{32}$$

where $L$ is the luminosity, $L^*$ is the characteristic luminosity and $\Psi^*$ is the normalization. The magnitude version is

$$\Psi(M; c, M^*, \Psi^*)dM = \frac{0.4 \, \Psi^* \, c^2 \ln(10) \, e^{-c10^{-0.4M+0.4M^*}} \left(10^{-0.4M+0.4M^*} + 10^{-0.8M+0.8M^*}\right)}{1+c}, \tag{33}$$

where $M$ is the absolute magnitude, $M^*$ the characteristic magnitude and $\Psi^*$ is the normalization. A test is performed on the $u^*$ band of the Sloan Digital Sky Survey (SDSS) as in [43] with data available at https://cosmo.nyu.edu/blanton/lf.html. The Schechter function, the new Lindley LF represented by Formula (33) and the data are reported in Figure 5, parameters as in Table 3.

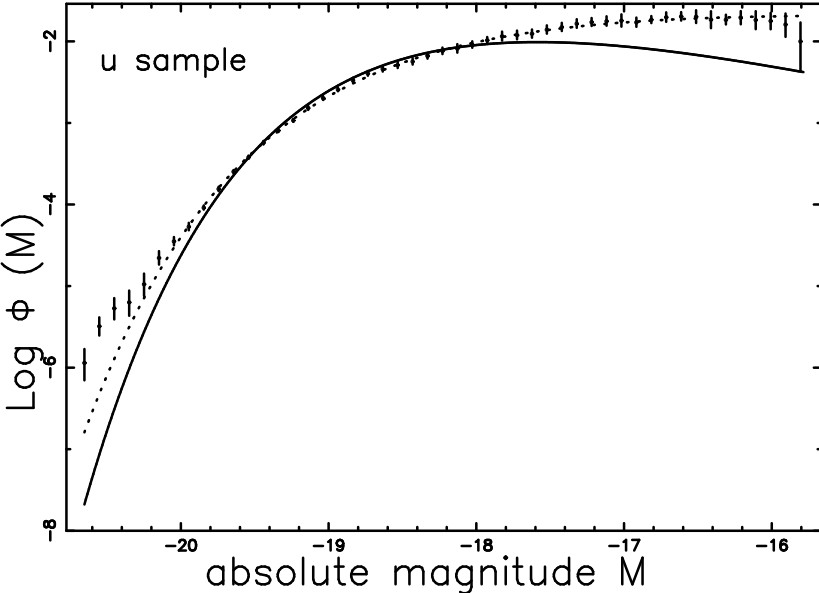

**Figure 5.** The luminosity function data of SDSS($u^*$) are represented with error bars. The continuous line fit represents the Lindley LF (33) and the dotted line represents the Schechter function.

**Table 3.** Numerical values values and $\chi^2_{red}$ of the LFs applied to SDSS Galaxies in the $u^*$ band.

| LF | Parameters | $\chi^2_{red}$ |
|---|---|---|
| Schechter | $M^* = -17.92$, $\alpha = -0.9$, $\Phi^* = 0.0114/\text{Mpc}^3$ | 0.689 |
| Lindley | $M^* = -23.40$, c = 214.1, $\Psi^* = 0.0289/\text{Mpc}^3$ | 6.6741 |
| truncated Lindley | $M^* = -23.458$; c = 224.47; $\Psi^* = 0.0239/\text{Mpc}^3$; $M_l = -20.653$; $M_u = -15.785$ | 6.6739 |

*4.3. The Truncated Lindley LF*

We start with the truncated Lindley PDF with scaling as given by Equation (20)

$$\Psi(L; c, L^*, \Psi^*, L_l, L_u)dL = \frac{\Psi^* \, \mathrm{e}^{-\frac{cL}{L^*}}\,(L+L^*)\,c^2}{DT},$$

(34)

with

$$DT = L^*\left(\mathrm{e}^{-\frac{cL_l}{L^*}}L^*c + \mathrm{e}^{-\frac{cL_l}{L^*}}cL_l - \mathrm{e}^{-\frac{cL_u}{L^*}}L^*c - \mathrm{e}^{-\frac{cL_u}{L^*}}cL_u + \mathrm{e}^{-\frac{cL_l}{L^*}}L^* - \mathrm{e}^{-\frac{cL_u}{L^*}}L^*\right),$$

(35)

where $L$ is the luminosity, $L^*$ is the characteristic luminosity, $L_l$ is the lower boundary in luminosity, $L_u$ is the upper boundary in luminosity, and $\Psi^*$ is the normalization. The magnitude version is

$$\Psi(M; c, M^*, \Psi^*, M_l, M_u)dM = \frac{NM}{DT}$$

(36)

where

$$NM = 0.4\,c^2\Psi^*\,(\ln(2) + \ln(5))\,\mathrm{e}^{c\left(10^{-0.4\,M_l + 0.4\,M^*} + 10^{0.4\,M^* - 0.4\,M_u} - 10^{0.4\,M^* - 0.4\,M}\right)} \times$$
$$\left(10^{0.4\,M_l + 0.4\,M_u}100^{0.4\,M^* - 0.4\,M} + 10^{0.4\,M_l + 0.4\,M^* + 0.4\,M_u - 0.4\,M}\right)$$

(37)

$$DT = 10^{0.4\,M_l + 0.4\,M^*}\mathrm{e}^{c10^{-0.4\,M_l + 0.4\,M^*}}c + \mathrm{e}^{c10^{-0.4\,M_l + 0.4\,M^*}}10^{0.4\,M_l + 0.4\,M_u}c$$
$$-10^{0.4\,M^* + 0.4\,M_u}\mathrm{e}^{c10^{0.4\,M^* - 0.4\,M_u}}c - \mathrm{e}^{c10^{0.4\,M^* - 0.4\,M_u}}10^{0.4\,M_l + 0.4\,M_u}c$$
$$+\mathrm{e}^{c10^{-0.4\,M_l + 0.4\,M^*}}10^{0.4\,M_l + 0.4\,M_u} - \mathrm{e}^{c10^{0.4\,M^* - 0.4\,M_u}}10^{0.4\,M_l + 0.4\,M_u},$$

(38)

where $M$ is the absolute magnitude, $M^*$ the characteristic magnitude, $M_l$ the lower boundary in magnitude, $M_u$ the upper boundary in magnitude and $\Psi^*$ is the normalization. The mean theoretical absolute magnitude, $\langle M \rangle$, can be evaluated as

$$\langle M \rangle = \frac{\int_{M_l}^{M_u} M \times \Psi(M; c, M^*, \Psi^*, M_l, M_u)dM}{\int_{M_l}^{M_u} \Psi(M; c, M^*, \Psi^*, M_l, M_u)dM}.$$

(39)

At the moment of writing, the analytical solution does not exists and the integration should be done numerically. Table 3 reports the parameters of the truncated Lindley LF from which is possible to conclude that the effect of truncation in the Lindley LF produces a minimum decrease in the $\chi^2_{red}$: Lindley LF with truncation $\chi^2_{red} = 6.6739$ and Lindley LF $\chi^2_{red} = 6.6741$.

*4.4. The Photometric Maximum*

In the pseudo-Euclidean universe, the correlation between expansion velocity and distance is

$$V = H_0 D = c_l\,z,$$

(40)

where $H_0$ is the Hubble constant, after [44], $H_0 = 100h$ km s$^{-1}$ Mpc$^{-1}$, with $h = 1$ when $h$ is not specified, $D$ is the distance in Mpc, $c_l$ is the light velocity and $z$ is the redshift. In the pseudo-Euclidean universe the flux of radiation, $f$, expressed in $\frac{L_\odot}{Mpc^2}$ units, where $L_\odot$ represents the luminosity of the sun, is

$$f = \frac{L}{4\pi D^2},$$ (41)

where $D$ represents the distance of the galaxy expressed in Mpc, and

$$D = \frac{c_l z}{H_0}.$$ (42)

The joint distribution in $z$ and $f$ for the Schechter LF, see formula (1.104) in [25] or formula (1.117) in [26], is

$$\frac{dN}{d\Omega dz df} = 4\pi \left(\frac{c_l}{H_0}\right)^5 z^4 \Phi\left(\frac{z^2}{z_{crit}^2}\right),$$ (43)

where $d\Omega$, $dz$ and $df$ represent the differential of the solid angle, the redshift and the flux respectively and $\Phi$ is the Schechter LF. The critical value of $z$, $z_{crit}$, is

$$z_{crit}^2 = \frac{H_0^2 L^*}{4\pi f c_l^2},$$ (44)

where $L^*$ has been defined in Section 4.1. The number of galaxies in $z$ and $f$ for the Schechter LF as given by formula (43) has a maximum at $z = z_{pos-max}$, where

$$z_{pos-max} = z_{crit}\sqrt{\alpha + 2},$$ (45)

which can be re-expressed as

$$z_{pos-max}(f) = \frac{\sqrt{2+\alpha}\sqrt{10^{0.4 M_\odot - 0.4 M^*}} H_0}{2\sqrt{\pi}\sqrt{f} c_l},$$ (46)

where $M_\odot$ is the reference magnitude of the sun at the considered bandpass. The position of the maximum in redshift for the Schechter LF depends from the flux of the selected astronomical band, $f$, and from the two parameter which characterizes the Schechter LF: $\alpha$ and $M^*$.

More details can be found in [45].

The joint distribution in $z$ and $f$ for galaxies for the Lindley LF, see Equation (34), is

$$\frac{dN}{d\Omega dz df} = \frac{4 z^4 c^2 e^{-\frac{c z^2}{z_{crit}^2}} c_l^5 \pi \left(z^2 + z_{crit}^2\right)}{(1+c) H_0^5 L^* z_{crit}^2}.$$ (47)

The maximum in the number of galaxies for the Lindley LF as function of $z_{crit}$ is at

$$z_{pos-max}(z_{crit}) = \frac{\sqrt{2}\sqrt{-c+3+\sqrt{c^2+2c+9}} z_{crit}}{2\sqrt{c}},$$ (48)

or as function of the flux $f$

$$z_{pos-max}(f) = \frac{\sqrt{2}\sqrt{-c+3+\sqrt{c^2+2c+9}}\sqrt{10^{0.4 M_\odot - 0.4 M^*}} H_0}{4\sqrt{c}\sqrt{\pi}\sqrt{f} c_l},$$ (49)

or as a function of the apparent magnitude $m$

$$z_{pos-max}(m) = \frac{5 \times 10^{-6} \sqrt{2} \sqrt{-c+3+\sqrt{c^2+2c+9}} \sqrt{10^{0.4M_{\odot}-0.4M^*}} H_0}{\sqrt{c} \sqrt{e^{0.921034M_{\odot}-0.921034m}} c_l}. \tag{50}$$

The position of the maximum in redshift for the Lindley LF depends from the flux of the selected astronomical band, $f$, or the selected apparent magnitude, $m$, and from the two parameter which characterizes the Lindley LF: $c$ and $M^*$.

The mean redshift for galaxies $\langle z \rangle$ can be defined as

$$\langle z \rangle = \frac{\int_0^{\infty} z \frac{dN}{d\Omega dz df} dz}{\int_0^{\infty} \frac{dN}{d\Omega dz df} dz}. \tag{51}$$

The mean redshift for the Lindley LF as function of $z_{crit}$ is

$$\langle z \rangle (z_{crit}) = \frac{16 z_{crit} (c+3)}{3 \sqrt{\pi} \sqrt{c} (2c+5)}, \tag{52}$$

or as a function of the flux

$$\langle z \rangle (f) = \frac{8 \sqrt{\pi f 10^{0.4M_{\odot}-0.4M^*}} H_0 (c+3)}{3 \pi^{3/2} f c_l \sqrt{c} (2c+5)} \tag{53}$$

or as a function of the apparent magnitude

$$\langle z \rangle (m) = \frac{3.009 \, 10^{-5} \sqrt{e^{0.921034M_{\odot}-0.921034m} 10^{0.4M_{\odot}-0.4M^*}} H_0 (c+3)}{e^{0.9210340374M_{\odot}-0.9210340374m} c_l \sqrt{c} (2c+5)}. \tag{54}$$

Figure 6 reports the number of observed galaxies of the 2MASS Redshift Survey (2MRS) catalog for a given apparent magnitude and the two theoretical curves are analyzed with same parameters as in Table 4. These parameters are derived in such a way that the $\chi^2$ is minimum. Therefore, this is a new method to derive the parameters which characterize the two LFs here adopted without using the samples for the LF such as the five bands of SDSS galaxies.

**Table 4.** Numerical values values and $\chi^2_{red}$ of the two LFs applied to $K_S$ band (2MASS Kron magnitudes) when $M_{\odot} = 3.39$.

| LF | Parameters | $\chi^2_{red}$ |
|---|---|---|
| Schechter | $M^* = -23.289$, $\alpha = -0.794$, $\Phi^* = 0.0128/\text{Mpc}^3$ | 7.08 |
| Lindley | $M^* = -23.7$, $c = 2.8$, $\Phi^* = 0.0289/\text{Mpc}^3$ | 6.84 |

### 4.5. Averaged Absolute Magnitude

We now introduce the concept of limiting apparent magnitude. The observable absolute magnitude as a function of the limiting apparent magnitude, $m_L$, is

$$M_L = m_L - 5 \log_{10} \left( \frac{cz}{H_0} \right) - 25. \tag{55}$$

Figure 7 presents such a curve and the galaxies of the 2MRS.

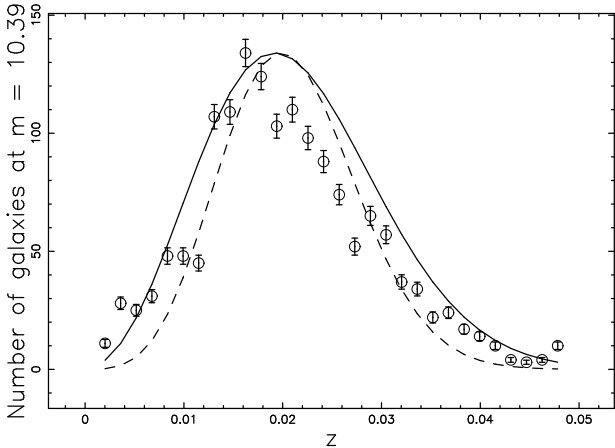

**Figure 6.** The galaxies of the 2MRS with $10.28 \leq m \leq 10.44$ or $1{,}202{,}409 \frac{L_\odot}{Mpc^2} \leq f \leq 1{,}384{,}350 \frac{L_\odot}{Mpc^2}$ are organized in frequencies versus heliocentric redshift, (empty circles); the error bar is given by the square root of the frequency. The maximum frequency of observed galaxies is at $z = 0.018$. The full line is the theoretical curve generated by $\frac{dN}{d\Omega dz df}(z)$ as given by the application of the Schechter LF which is Equation (43) and the dashed line represents the Lindley LF which is Equation (47). The parameters are the same of Table 4, $\chi^2 = 198$ for the Schechter LF and $\chi^2 = 191$ for the Lindley LF.

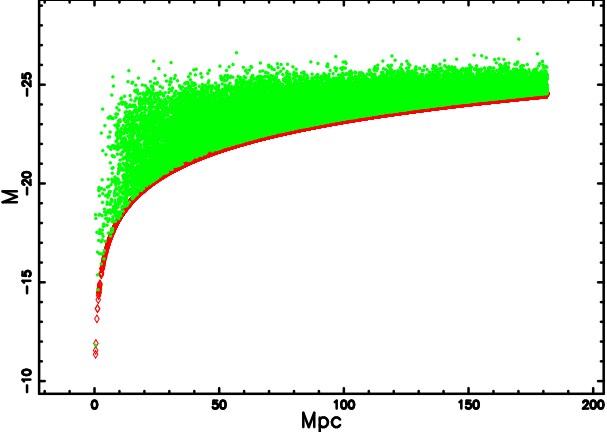

**Figure 7.** The absolute magnitude $M$ of 36,474 galaxies belonging to the 2MRS when $\mathcal{M}_\odot = 3.39$ and $H_0 = 70$ km s$^{-1}$ Mpc$^{-1}$ (green points). The lower theoretical curve as represented by Equation (55) is shown as the red-thick line when $m_L = 11.75$.

We now compare the theoretical averaged absolute magnitude of the truncated Lindley LF, see Equation (39), with the observed averaged absolute magnitude of 2MRS as function of the redshift. To fit the data we assumed the following empirical dependence with redshift for the characteristic magnitude of the truncated Lindley LF

$$M^* = -25.14 + 3\left(1 - \sqrt{\frac{z - z_{min}}{z_{max} - z_{min}}}\right). \tag{56}$$

This relationship models the decrease of the characteristic absolute magnitude as function of the redshift and allows us to match observational and theoretical data. The lower bound in absolute magnitude is given by the minimum magnitude of the selected bin, the upper bound is given by Equation (55), the characteristic magnitude varies according to Equation (56) and Figure 8 reports the comparison between theoretical and observed absolute magnitude for 2MRS.

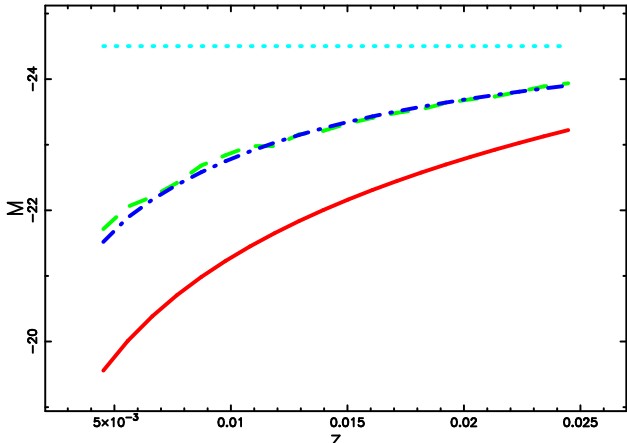

**Figure 8.** Averaged absolute magnitude of the galaxies belonging to the 2MRS (green-dashed line), theoretical averaged absolute magnitude for the truncated Lindley LF (blue dash-dot-dash-dot line) as given by Equation (39), lower theoretical curve as represented by Equation (55) (red line) and minimum absolute magnitude observed (cyan dotted line).

## 5. Conclusions

**Statistical Distributions.** We introduced the Lindley distribution with scale and the truncated Lindley distribution. The parameters of the Lindley distribution with scale can be found with the method of the matching moments. In the the case of the truncated Lindley distribution the MLE is used to estimate the unknown parameters.

**Application to the stars.** To fit the IMF for stars with the truncated Lindley PDF, the parameters $b$ and $c$, which is deduced from the astronomical sample, should be positive. This is the case of NGC 6611 (207 stars + BDs), for which the reduced merit function is smaller for the truncated Lindley distribution in respect to the lognormal and truncated lognormal distribution, see Table 1.

**Application to the galaxies.** The Lindley LF for galaxies is characterized by a higher reduced merit function in respect to the Schechter LF for the case of SDSS Galaxies in the $u^*$ band, see Table 3. Conversely the Lindley LF for galaxies produces a lower value of the merit function when the photometric maximum of 2MRS is modeled in respect to the Schechter model for the maximum, see Figure 6. The truncated Lindley LF produces an acceptable model for the averaged absolute magnitude of the galaxies belonging to the 2MRS, see Figure 8.

**Funding:** This research received no external funding.

**Conflicts of Interest:** The authors declare no conflict of interest.

## Appendix A. Other Parameters of the Lindley Distribution

The $r$th moment about the origin for the Lindley distribution is, $\mu'_r$, is

$$\mu'_r = \frac{c^{-r}\Gamma\,(r+2) + c^{1-r}\Gamma\,(r+1)}{1+c}, \tag{A1}$$

where

$$\Gamma(z) = \int_0^{\infty} e^{-t}t^{z-1}dt, \tag{A2}$$

is the gamma function, see [30]. The central moments, $\mu_r$, are

$$\mu_3 = \frac{2\,c^3 + 12\,c^2 + 12\,c + 4}{c^3\,(1+c)^3} \tag{A3a}$$

$$\mu_4 = \frac{9\,c^4 + 72\,c^3 + 132\,c^2 + 96\,c + 24}{c^4\,(1+c)^4} \tag{A3b}$$

being $\mu_2 = \sigma^2$. Is impossible to evaluate the median in a closed form and therefore we introduce an approximated distribution function, $F_{2,2}$ in terms of the Padé rational polynomial approximation, after [46], of degree 2 in the numerator and degree 2 in the denominator about the point $x = 0$

$$F_{2,2} = \frac{6\,xc^2\,\left(2\,c^2 - cx - 4\,c + 3\,x + 6\right)}{\left(c^4x^2 - 4\,c^3x^2 + 6\,c^3x + 7\,c^2x^2 - 18\,c^2x + 12\,c^2 + 24\,cx - 24\,c + 36\right)\,(1+c)} \tag{A4}$$

The approximated median, $m_{2,2}$ turns out to be

$$m_{2,2} = \frac{9\,c^3 - 18\,c^2 - \sqrt{69\,c^6 - 276\,c^5 + 690\,c^4 - 876\,c^3 + 1101\,c^2 - 984\,c + 1188} + 33\,c - 12}{\left(c^3 - 3\,c^2 + 15\,c - 29\right)c}. \tag{A5}$$

The percent error, $\delta$, in the evaluation of the approximated median is $\delta = 1.179\,\%$ at c = 0.5 and $\delta = 0.077\,\%$ at c = 2.

## Appendix B. Moments for the Lindley Distribution with Scale

The $r$th moment about the origin for the Lindley distribution with scale, $\mu'_{r,s}$, is

$$\mu'_{r,s} = \frac{c^{-r}b^r\Gamma\,(r+2)}{1+c} + \frac{c^{1-r}b^r\Gamma\,(r+1)}{1+c}. \tag{A6}$$

The central moments, $\mu_{r,s}$, are

$$\mu_{3,s} = \frac{2\,b^3\,\left(c^3 + 6\,c^2 + 6\,c + 2\right)}{c^3\,(c^3 + 3\,c^2 + 3\,c + 1)} \tag{A7a}$$

$$\mu_{4,s} = \frac{3\,b^4\,\left(3\,c^4 + 24\,c^3 + 44\,c^2 + 32\,c + 8\right)}{c^4\,(c^4 + 4\,c^3 + 6\,c^2 + 4\,c + 1)}. \tag{A7b}$$

## Appendix C. The Parameters of the Truncated Lindley Distribution

The parameters of the truncated Lindley distribution can be obtained from empirical data by the maximum likelihood estimators (MLE) and by the evaluation of the minimum and maximum elements of the sample. Consider a sample $\mathcal{X} = x_1, x_2, \ldots, x_n$ and let $x_{(1)} \geq x_{(2)} \geq \cdots \geq x_{(n)}$ denote their order statistics, so that $x_{(1)} = \max(x_1, x_2, \ldots, x_n)$, $x_{(n)} = \min(x_1, x_2, \ldots, x_n)$. The first two parameters $x_l$ and $x_u$ are

$$x_l = x_{(n)}, \qquad x_u = x_{(1)}. \tag{A8}$$

The MLE is obtained by maximizing

$$\Lambda = \sum_i^n \ln(f_t(x_i; b, c, x_l, x_u)). \tag{A9}$$

The two derivatives $\frac{\partial\Lambda}{\partial b} = 0$ and $\frac{\partial\Lambda}{\partial c} = 0$ generate two non-linear equations in $b$ and $c$ which can be solved numerically, we used FORTRAN subroutine SNSQE in [47],

$$\frac{\partial \Lambda}{\partial b} = \frac{PNB}{b^2 \left( -\mathrm{e}^{-\frac{cx_u}{b}} bc - \mathrm{e}^{-\frac{cx_u}{b}} cx_u + \mathrm{e}^{-\frac{cx_l}{b}} bc + \mathrm{e}^{-\frac{cx_l}{b}} cx_l - \mathrm{e}^{-\frac{cx_u}{b}} b + \mathrm{e}^{-\frac{cx_l}{b}} b \right)} = 0, \tag{A10}$$

where

$$PNB = -\mathrm{e}^{-\frac{cx_u}{b}} bc^2 nx_u - \mathrm{e}^{-\frac{cx_u}{b}} c^2 nx_u{}^2 + \mathrm{e}^{-\frac{cx_l}{b}} bc^2 nx_l + \mathrm{e}^{-\frac{cx_l}{b}} c^2 nx_l{}^2 - 2\,\mathrm{e}^{-\frac{cx_u}{b}} b^2 cn$$

$$-2\,\mathrm{e}^{-\frac{cx_u}{b}} bcnx_u + 2\,\mathrm{e}^{-\frac{cx_l}{b}} b^2 cn + 2\,\mathrm{e}^{-\frac{cx_l}{b}} bcnx_l + \sum_{i=1}^{n} \frac{cx_i b + cx_i{}^2 + b^2}{x_i + b} \mathrm{e}^{-\frac{cx_u}{b}} bc$$

$$+ \sum_{i=1}^{n} \frac{cx_i b + cx_i{}^2 + b^2}{x_i + b} \mathrm{e}^{-\frac{cx_u}{b}} cx_u - \sum_{i=1}^{n} \frac{cx_i b + cx_i{}^2 + b^2}{x_i + b} \mathrm{e}^{-\frac{cx_l}{b}} bc$$

$$- \sum_{i=1}^{n} \frac{cx_i b + cx_i{}^2 + b^2}{x_i + b} \mathrm{e}^{-\frac{cx_l}{b}} cx_l - 2\,\mathrm{e}^{-\frac{cx_u}{b}} b^2 n + 2\,\mathrm{e}^{-\frac{cx_l}{b}} b^2 n + \sum_{i=1}^{n} \frac{cx_i b + cx_i{}^2 + b^2}{x_i + b} \mathrm{e}^{-\frac{cx_u}{b}} b$$

$$- \sum_{i=1}^{n} \frac{cx_i b + cx_i{}^2 + b^2}{x_i + b} \mathrm{e}^{-\frac{cx_l}{b}} b. \tag{A11}$$

and

$$\frac{\partial \Lambda}{\partial c} = \frac{PNC}{cb \left( -\mathrm{e}^{-\frac{cx_u}{b}} bc - \mathrm{e}^{-\frac{cx_u}{b}} cx_u + \mathrm{e}^{-\frac{cx_l}{b}} bc + \mathrm{e}^{-\frac{cx_l}{b}} cx_l - \mathrm{e}^{-\frac{cx_u}{b}} b + \mathrm{e}^{-\frac{cx_l}{b}} b \right)} = 0, \tag{A12}$$

where

$$PNC = -\mathrm{e}^{-\frac{cx_u}{b}} bc^2 nx_u - \mathrm{e}^{-\frac{cx_u}{b}} c^2 nx_u{}^2 + \mathrm{e}^{-\frac{cx_l}{b}} bc^2 nx_l + \mathrm{e}^{-\frac{cx_l}{b}} c^2 nx_l{}^2 + \mathrm{e}^{-\frac{cx_u}{b}} \sum_{i=1}^{n} x_i bc^2$$

$$+ \mathrm{e}^{-\frac{cx_u}{b}} \sum_{i=1}^{n} x_i c^2 x_u - \mathrm{e}^{-\frac{cx_u}{b}} b^2 cn - 2\,\mathrm{e}^{-\frac{cx_u}{b}} bcnx_u - \mathrm{e}^{-\frac{cx_l}{b}} \sum_{i=1}^{n} x_i bc^2 - \mathrm{e}^{-\frac{cx_l}{b}} \sum_{i=1}^{n} x_i c^2 x_l$$

$$+ \mathrm{e}^{-\frac{cx_l}{b}} b^2 cn + 2\,\mathrm{e}^{-\frac{cx_l}{b}} bcnx_l + \mathrm{e}^{-\frac{cx_u}{b}} \sum_{i=1}^{n} x_i bc - 2\,\mathrm{e}^{-\frac{cx_u}{b}} b^2 n$$

$$- \mathrm{e}^{-\frac{cx_l}{b}} \sum_{i=1}^{n} x_i bc + 2\,\mathrm{e}^{-\frac{cx_l}{b}} b^2 n. \tag{A13}$$

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
