# Peer review of "The Truncated Lindley Distribution with Applications in Astrophysics"

_galaxies, doi:10.3390/galaxies7020061_

Reviewer 1 Report

The author study the Lindsley distribution (LD) and some of its variants and applies them to different areas of astrophysics: the initial mass function of stars, the luminosity function of SDSS galaxies, the photometric maximum of 2MRS galaxies, and the Malmquist bias of these galaxies.

While I have not gone through the (numerous) equations in details, the results seem correct, but of limited interest to the community.  In addition, several points should be clarified before the paper can be considered for publication in Galaxies

Major comments:

- The paper has a lot of equations and calculations, and is repetitive (Sections 2,3, and 4 are the same calculations applied to three different cases: the Lindley distribution, the LD with scale, and the truncated LD with scale). It would be more clear to summarize the key points (definition, mean, variance, and a few other key quantities) into one section, and defer the other calculations to the Appendix. While the LD itself certainly deserves its own paragraph, the other two could be introduced together in a second subsection of the LD section as they are very similar.

- Most of the times, the results are stated with no comment or discussion. It would add to the paper to interpret the results and explain why the LD is interesting rather than just stating best fit results.

- The LD does not seem to do as well as the Schechter function, and has no theoretical backing, therefore it is not clear whether it is useful to show it here, although, admittedly a negative result is still a result. 

- How is it that the LD does not fit the galaxy luminosity distribution as well as the Schechter one, while for the photometric maxium case, it works better?

- what is the justification for eq. 61?

Minor comments : 

- Fig. 4  takes the whole page, and its purpose is not clear. 

- Eqs. (37–39) have a strange (black square character), is it meant to be Psi*?  

- Malmquist bias usually referts the the preferential detection of brighter sources at high redshift. Here, it seems to simply mean the mean brightness as a function of redshift. I think it would be good to clarify, and if necessary, remove the term "Malmquist bias"

Author Response

Here you will find the answer organized by points.

Point of the referee 1)
  The paper has a lot of equations and calculations, and is repetitive
 (Sections 2,3, and 4 are the same calculations applied to three different
 cases: the Lindley distribution, the LD with scale, and the truncated LD
 with scale). It would be more clear to summarize the key points
  (definition, mean, variance, and a few other key quantities) into one
  section, and defer the other calculations to the Appendix. While the LD
  itself certainly deserves its own paragraph, the other two could be
 introduced together in a second subsection of the LD section as they are
 very similar.
Answer of the author to point 1)
  The Lindley family is now concentrated  in Section 2 as suggested.
  Some results are reported in two appendices as suggested.

Point of the referee 2)
- Most of the times, the results are stated with no comment or discussion.
Answer of the author to point 2)
  *) An example  of the utility for the  truncated IMF has been
     reported  before Section 4.
  *) A comment has been inserted after equation (31)
     on M^*

Point of the referee 3)
- It would add to the paper to interpret the results and explain why the LD
- is interesting rather than just stating best fit results.
Answer of the author to point 3)
  As an example  now the problem of the IMF is better introduced
  , see beginning of Section 3.
  The truncated Lindley distribution has been introduced
  because there is maximum and a minimum in the IMF for stars,
  see text  in  Section 3.

Point of the referee 4)
- The LD does not seem to do as well as the Schechter function, and has no
- theoretical backing, therefore it is not clear whether it is useful to
- show it here, although, admittedly a negative result is still a result.
- How is it that the LD does not fit the galaxy luminosity distribution as
- well as the Schechter one, while for the photometric maxium case, it works
- better?
Answer of the author to point 4)
  a) the target  for a new LF for galaxies has been summarized at the
     end of  section 4.1.
  b) the parameters for the two LFs here used in the
     case of the  photometric maximum are chosen minimizig
     the chisquare, see new text after formula (54)

Point of the referee 5)
- what is the justification for eq. 61?
Answer of the author to point 5)
  Old formula (61) now (56) allows to match observational
  and theoretical data. This comment has  been inserted
  after equation (56).

Point of the referee 6)
- Fig. 4  takes the whole page, and its purpose is not clear.
Answer of the author to point 6)
  a) a new figure has been inserted
  b) an evaluation of the reduction in the number of stars
     after a given time is given ,
     see new text before section 4.

Point of the referee 7)
- Eqs. (3739) have a strange (black square character), is it meant to be
- Psi*?
Answer of the author to point 7)
  The strange  black square character in Eqs. (3739)
  now equations (3437) has been introduced
  by the Galaxies version of the pdf.
  I will see how to cook this fact in the future with Galaxies.

Point of the referee 8)
- Malmquist bias usually referts the the preferential detection of brighter
- sources at high redshift. Here, it seems to simply mean the mean
- brightness as a function of redshift. I think it would be good to clarify,
- and if necessary, remove the term "Malmquist bias"
Answer of the author to point 8)
  I have adopted the point of view of the referee and I am now
  speaking of averaged absolute magnitude.
  The term Malmquist bias has been removed.

Reviewer 2 Report

Introduction

The introduction part should be improved in order to introduce more about the background of related astrophysics and explain why the Lindley distribution is valuable/necessary  to be applied to the astrophysics. This is a paper on Astrophysics & astronomy, but after I read the introduction part I find the necessary scientific background for astrophysics is absent.    

Section 2 & 3.
The author listed many basic equations, which can be deduced easily. Are all these equations necessary to be listed in the paper? I suggest the author just list the important and necessary equations.  

Section 4.
Considering that the truncated Lindley distribution is the key point in this paper, I suggest that the author give some common on the  truncated Lindley distribution and discuss its advantages.

Section 5.
The author should give a brief introduction about the background of the IMF of stars and explain why the  truncated Lindley distribution is a preferable choice.

Section 6.
 Equation (37) : What does the solid square mean? Is it a free parameter? Maybe this is not a good choice to  use such symbol to represent a parameter.

Overall, I think the paper put too many mathematical formulas, which make the scientific concentration seems to be flooded. I think the paper should be improved greatly in order to highlight the scientific theme.

Author Response

Point of the referee 1)
 Introduction
The introduction part should be improved in order to introduce more about
the background of related astrophysics and explain why the Lindley
distribution is valuable/necessary  to be applied to the astrophysics. This
is a paper on Astrophysics & astronomy, but after I read the introduction
part I find the necessary scientific background for astrophysics is absent.
Answer of the author to point 1)
Now  the introduction  contains a detailed discussion on
the IMF for stars,
the LF for galaxies,
the photometric maximum for galaxies and
the range in absolute magnitude for galaxies
versus the redshift.
Point of the referee 2)
Section 2 & 3.
The author listed many basic equations, which can be deduced easily. Are all
these equations necessary to be listed in the paper? I suggest the author
just list the important and necessary equations.
Answer of the author to point 2)
The non necessary equations are now inserted in appendix A and B

Point of the referee 3)
Section 4.
Considering that the truncated Lindley distribution is the key point in this
paper, I suggest that the author give some common on the  truncated Lindley
distribution and discuss its advantages.
Answer of the author to point 3)
The introduction now contains a more clear discussion on the
need to substitute the zero and infinity in the PDFs
with finite values.
The truncated Lindley PDF was not yet analyzed in astrophysics.

Point of the referee 4)
Section 5.
The author should give a brief introduction about the background of the IMF
of stars and explain why the  truncated Lindley distribution is a preferable
choice.
Answer of the author to point 4)
 a) The problem of the IMF is now better introduced
    in two places : the introduction
    and the  beginning of Section 3.
  The truncated Lindley distribution has been introduced
  because there is maximum and a minimum in the IMF for stars,
  see text  between (28-29)  in  Section 3.

Point of the referee 5)
Section 6.
 Equation (37) : What does the solid square mean? Is it a free parameter?
Maybe this is not a good choice to  use such symbol to represent a
parameter.
Answer of the author to point 5)
  The strange  black square character in old equation (37)
  now equation  (32)  has been introduced
  in  the Galaxies version of the pdf.
  I will see how to cook this fact in the future with the Galaxies
  redaction.

Round  2

Reviewer 1 Report

Most of the points have been satisfactorily answered, and I find the manuscript more clear, although the motivations and the usefulness of this family of distributions are still not completely convincing to me. 

I only have one last comment: Fig. 5 compares the Schechter and Lindley LFs, and clearly, the Lindley LF is not a good fit. Then in section 4.3, the author calculates the truncated Lindley LF (TLLF). If this LF is meant to be a better fit than Schechter, why is it not plotted on Fig. 5, and the best-fit values should be reported (compared with table 3). 

If the fit is not better, then I don't see the point of the TLLF in this section, since the rest of section 4 uses the non-truncated Lindley LF.

Author Response

Point of the referee 1
I only have one last comment:
Fig. 5 compares the Schechter and Lindley LFs,
and clearly, the Lindley LF is not a good fit.
Then in section 4.3, the
author calculates the truncated Lindley LF (TLLF).
If this LF is meant to be
a better fit than Schechter,
why is it not plotted on Fig. 5, and the
best-fit values should be reported (compared with table 3).
If the fit is not better, then I don't see the point of
the TLLF in this
section, since the rest of section 4 uses the non-truncated Lindley LF.

Answer to point 1 of the author

Now table 3 contains the parameters of the truncated Lindley LF.
The differences between  truncated and not truncated Lindley
LF are now outlined , see red text before  Section 3.3.

Reviewer 2 Report

Section 3: The author shoud give at least a  brief discussion on his result. E.g., Discuss the physical meaning, compare the Lindley method with traditional ones and thus highlight the strengths of the Lindley method.

Section 4: This section has the same problem. The author should give a more detailed discussion on the result. The author have a good mathematical background and present much mathematical description, however the discussion on aspect of astronomy is deficient.

Section 5: This setion should  be adjusted once Section 3 and 4 are revised.

Author Response

Points of the referee

Section 3: The author shoud give at least a
brief discussion on his result.
E.g., Discuss the physical meaning,
compare the Lindley method with
traditional ones and thus highlight the strengths of
the Lindley method.

Section 4: This section has the same problem.
The author should give a more
detailed discussion on the result.
The author have a good mathematical
background and present much mathematical description,
however the discussion
on aspect of astronomy is deficient.

Section 5: This setion should  be adjusted
once Section 3 and 4 are revised.

Answer of the author

I have introduced five modifications to the text as suggested by
the referee. They are marked in red.